# Microalgae to Bioenergy: Optimization of *Aurantiochytrium* sp. Saccharification

**DOI:** 10.3390/biology12070935

**Published:** 2023-06-29

**Authors:** Joana Oliveira, Sara Pardilhó, Joana M. Dias, José C. M. Pires

**Affiliations:** 1Laboratory for Process Engineering, Environment, Biotechnology and Energy (LEPABE), Department of Chemical Engineering, Faculty of Engineering, University of Porto, 4200-465 Porto, Portugal; up201904660@edu.fe.up.pt; 2ALiCE—Associate Laboratory in Chemical Engineering, Faculty of Engineering, University of Porto, 4200-465 Porto, Portugal; spardilho@fe.up.pt; 3Laboratory for Process Engineering, Environment, Biotechnology and Energy (LEPABE), Department of Metallurgical and Materials Engineering, Faculty of Engineering, University of Porto, 4200-465 Porto, Portugal

**Keywords:** thermal acid hydrolysis, microalgae, *Aurantiochytrium* sp., response surface methodology, bioethanol production

## Abstract

**Simple Summary:**

Over recent years, microalgae have gained attention as a potential source for bioethanol production due to their fast growth rates, ability to grow in several environments and carbohydrate content, mainly in starch form. Biomass pretreatment is considered a critical step in bioethanol production since fermentable sugars are required for yeast growth. Although many studies have been conducted on the thermal acid hydrolysis of microalgae, the use of systematic methodologies, such as the response surface methodology, to obtain a comprehensive view of the influence of the studied parameters is relatively unexplored. This study evaluates the influence of acid concentration, hydrolysis time and biomass/acid ratio on the thermal acid hydrolysis of *Aurantiochytrium* sp. through the response surface methodology, aiming at further bioethanol production. For each of the developed models, the maximum sugar concentration and yield were 18.05 g/L and 12.86 g/100 g, respectively. This study provides insights into the hydrolysis of microalgae for bioethanol production and suggests that *Aurantiochytrium* sp. microalgae could be a promising feedstock for bioethanol production.

**Abstract:**

Microalgae are a promising feedstock for bioethanol production, essentially due to their high growth rates and absence of lignin. Hydrolysis—where the monosaccharides are released for further fermentation—is considered a critical step, and its optimization is advised for each raw material. The present study focuses on the thermal acid hydrolysis (with sulfuric acid) of *Aurantiochytrium* sp. through a response surface methodology (RSM), studying the effect of acid concentration, hydrolysis time and biomass/acid ratio on both sugar concentration of the hydrolysate and biomass conversion yield. Preliminary studies allowed to establish the range of the variables to be optimized. The obtained models predicted a maximum sugar concentration (18.05 g/L; R^2^ = 0.990) after 90 min of hydrolysis, using 15% (*w*/*v*) biomass/acid ratio and sulfuric acid at 3.5% (*v*/*v*), whereas the maximum conversion yield (12.86 g/100 g; R^2^ = 0.876) was obtained using 9.3% (*w*/*v*) biomass/acid ratio, maintaining the other parameters. Model outputs indicate that the biomass/acid ratio and time are the most influential parameters on the sugar concentration and yield models, respectively. The study allowed to obtain a predictive model that is very well adjusted to the experimental data to find the best saccharification conditions for the *Aurantiochytrium* sp. microalgae.

## 1. Introduction

The increasing concerns over the use of fossil fuels reinforce the need to explore renewable alternatives to change the paradigm regarding the current energy crisis, such as the production of liquid biofuels [1,2]. Bioethanol is a renewable fuel produced through the fermentation of biomass [3]. The use of this biofuel is aligned with the United Nations Sustainable Development Goals, namely Goals 7 (Affordable and Clean Energy) and 13 (Climate Action), since it (i) can be produced from renewable sources, (ii) reduces net CO_2_ emissions—as the production of the feedstock usually involves CO_2_ absorption and (iii) increases energy security, since this fuel can be produced locally [1,2,4,5].

The most used feedstocks for bioethanol production are food crops (e.g., sugarcane, corn and wheat) as they are rich in fermentable sugars [1]. However, their use leads to competition with food production and arable land [6]; thus, alternative feedstocks such as residual lignocellulosic biomass have recently been used to produce bioethanol. However, high lignin content requires additional pretreatments to ferment the biomass, which are generally associated with high processing costs, making the search for alternative non-food raw materials relevant [7].

Over recent years, microalgae have gained attention as a potential source for bioethanol production due to their fast growth rates and ability to grow in various environments, including wastewater and seawater [2]. These microorganisms have a short harvesting cycle (approximately 1–10 d) [8], reaching high biomass productivity. Furthermore, the absence of lignin and its high carbohydrate content (which can reach 60 wt.%) indicate that such biomass can be easily converted into fermentable monosaccharides [7,9,10,11,12]. In particular, *Aurantiochytrium* sp. shows a carbohydrate content between 20 wt.% and 60 wt.% [7,13,14], depending on the species and growth conditions, which explains the interest in its use for bioethanol production.

Microalgae contain complex carbohydrates such as starch and cellulose and simpler sugars such as glucose, fructose and sucrose. Depending on the microalgal species and growth conditions, the carbohydrate content and composition can vary significantly [15]; therefore, the optimal hydrolysis conditions depend on the carbohydrate profile of each feedstock. No studies in the literature related to *Aurantiochytrium* sp. as a feedstock for bioethanol production or the optimization of saccharification were found.

Bioethanol production generally comprises three steps: (i) biomass pretreatment, (ii) alcoholic fermentation and (iii) bioethanol purification. The pretreatment step involves breaking down the complex carbohydrates into simple sugars that can be fermented by yeast [7]. This can be achieved through acid, alkaline or biological pretreatment [5]. Even though enzymatic pretreatment is highly specific and does not require harsh conditions, the high enzyme cost makes it unattractive to implement at an industrial scale [3,12]. On the other hand, the chemical process is associated with significantly lower costs [16]. In this case, after the pretreatment, the pH of the solution needs to be adjusted to 4.5–5.0 before adding the yeast and performing fermentation (e.g. with *Saccharomyces cerevisiae*) [6].

Biomass pretreatment is considered a critical step in bioethanol production since fermentable sugars are required for yeast growth [10]. Regarding chemical hydrolysis, acid pretreatment is most commonly used compared with alkali, leading to higher hydrolysis efficiency [8]. In that sense, biomass hydrolysis with dilute acid solutions at high temperatures has been demonstrated to be a simple and low-cost method to release those sugars without compromising the economic feasibility of the process [9,17].

An extensive literature has been published concerning the dilute thermal acid hydrolysis of microalgae for bioethanol production [1,2,6,8,9,11,16,17,18,19]. Miranda et al. [17] studied the use of *Scenedesmus obliquus* microalgae for bioethanol production, achieving the best results through acid hydrolysis with H_2_SO_4_ (2.8% (*v*/*v*)) and 5% (*w*/*v*) biomass concentration at 120 °C (autoclave) for 30 min leading to a sugar yield of 28.6 g_glucose equivalent_ per 100 g dry algae and a sugar concentration of 14.4 g/L. For the sugar concentration, the optimized conditions obtained included a much higher biomass concentration—50% (*w*/*v*).

Castro et al. [18] used a full factorial experiment to optimize saccharification conditions for bioethanol production of a mixture of microalgae species (*Scenedesmus*, *Chlorella*, *Ankistrosdemus*, *Micromonas* and *Chlamydomonas*) by dilute acid hydrolysis. The authors found that the optimal conditions were: 10% (*w*/*v*) solid loading, 2.8% (*v*/*v*) H_2_SO_4_ solution and 120 min hydrolysis at 80–90 °C, which led to a sugar yield of 16.61 g/100 g dry algae.

The response surface methodology (RSM) is a statistical technique used to optimize a process considering the modeling of the influence of several input variables on a selected response. This tool lowers the number of experiments necessary and makes reliable predictions [20]. No studies were found regarding the study of microalgae saccharification using the RSM. Therefore, the aim of this work was to gain a comprehensive view of how thermal acid hydrolysis reaction conditions (time, acid concentration and amount of biomass) influence the saccharification of *Aurantiochytrium* sp. biomass, considering the use of the RSM to model their impact, as response variables, on both sugar concentration of the hydrolysates and sugar yield, keeping in mind further bioethanol production.

## 2. Materials and Methods

### 2.1. Preliminary Studies: Definition of Variable Ranges

*Aurantiochytrium* sp. microalgal biomass was provided by Allmicroalgae—Natural Products S.A. Cultivation. The biomass was obtained from industrial-scale fermentation in a heterotrophic industrial medium with control of sugar content during fermentation. Fermentation conditions favored lipid accumulation (the main product to obtain with this microalga). Preliminary studies were carried out in duplicate to define the range of parameters (hydrolysis time, biomass/acid ratio and acid concentration) to be used in further optimization of the hydrolysis process. The reactions were conducted in a water bath (~90 °C) in 250 mL flasks using 100 mL H_2_SO_4_ solution (concentration ranging 2.5–4% (*v*/*v*)), with different biomass/acid ratios (1, 2.5 and 10% (*w*/*v*)) and hydrolysis times (30, 60 and 90 min). Based on the literature review [6,9,21], it was not considered relevant to perform preliminary studies at temperatures lower than 90 °C. Preliminary assays were conducted to evaluate the influence of the temperature in the sugar release in harsher conditions, namely with an autoclave (121 °C; Uniclave 88, AJC, Cacém, Portugal) and a thermoreactor (100 °C for 60 min plus 148 °C for 30 min). The other parameters, kept in the range previously mentioned, were selected based on the results obtained.

### 2.2. Experimental Design

For the optimization process, different hydrolysis times, biomass/acid ratios and H_2_SO_4_ concentrations were used in the thermal acid hydrolysis (water bath at ~90 °C) considering results obtained in the preliminary studies. Since the RSM usually involves the study of the behavior of relevant parameters to reduce the number of experiments that need to be conducted, the number of factors considered was 3, namely time, biomass/acid ratio and acid concentration, based on the preliminary experiments.

A central composite design (CCD; Face Centered) was used to evaluate the influence of the studied variables on sugar concentration and yield through JMP (SAS Institute; Cary, NC, USA) and Minitab (Minitab Inc., State College, PA, USA) software. Being the most commonly used response surface design of experiments, this methodology allows second-order polynomial models to be built [22]. The use of other standard designs for second-order models, such as the Box–Behnken design (BBD), was considered. Despite requiring fewer experiments and accurately estimating main effects and two-factor interactions, the BBD shows poorer performance in estimating quadratic effects, since it is not possible to check the adequacy of the quadratic model [23] and therefore was not chosen for the current work. Furthermore, Czyrski et al. [24] and Zolgharnein et al. [25] compared the use of the CCD, BBD and Doehlert design in the RSM and found that the CCD showed the best-fitting results. The three factors were studied at three levels, coded as −1, 0 and 1, with the ranges reported in Table 1. The independent variables were defined as *x*_1_ (time), *x*_2_ (biomass/acid ratio) and *x*_3_ (acid concentration).

The experimental data were analyzed to fit a second-order polynomial model (Equation (1)), considering the linear, quadratic and interaction effects of the exploratory variables. Two models were developed, considering the sugar concentration (*y*_1_) and the yield (*y*_2_) as the dependent variables.
(1)$$y=\beta_{0}+\sum_{\;}^{\;}{\;\beta }_{i}x_{i}+\sum_{\;}^{\;}{\;\beta }_{ii}x_{i}^{2}+\sum_{\;}^{\;}{\;\beta }_{ij}x_{i}x_{j}$$
where *y* is the predicted response; *β*_0_, *β_i_, β_ii_* and *β_ij_* are regression coefficients; and *i* and *j* are indexes of the number variables, where *i*, *j* ∈ {1,2,3} and *i* ≠ *j*.

Statistical analysis was performed to validate the model results. In that sense, the obtained models were evaluated through analysis of variance using Minitab. The models’ parameters were analyzed considering the *p*-value obtained in the analysis of variance and were considered statistically significant at a 95% confidence level. The variables with no statistical significance (*p* > 0.05) were removed in descending order of *p* in a stepwise algorithm. In that sense, the achieved models considered only statistically significant parameters (*p* < 0.05). The adjusted model and the respective coefficients of determination (R^2^) were used to evaluate the model performance in training and test sets. The training set was used to determine the model parameters, while the test set was used to validate the model (evaluation of the model predictive performance with new observations). In detail, R^2^ provides information regarding the fraction of the variance of the output variable explained by the model [26], whereas the predicted and adjusted R^2^ values are related to the performance of the model in predicting new data (test set) and provide a more conservative measure of model fit that avoids overfitting, since it takes into consideration the number of significant elements in the model [27,28]. The lack of fit and the analysis of residues were also considered to evaluate model performance.

JMP software was used to define the critical points of the models. Microsoft Excel Solver was used to determine the conditions where the responses were maximized.

Considering the obtained results, a hydrolysis assay with conditions corresponding to the maximum output value (namely 90 min; 10% (*w*/*v*) biomass/acid ratio and 3.5% (*v*/*v*) acid concentration) was performed to validate the achieved models. The experimental value was then compared with the model given by the model, and the predictive error was calculated.

### 2.3. Sugars Quantification

The samples were hydrolyzed and cooled to room temperature, and the supernatant was centrifuged at 4000 rpm for 10 min (HERMLE Z 200 A). The total sugar content was determined spectrophotometrically (UV mini-1240, UV–Vis spectrophotometer, Shimadzu, Kyoto, Japan) at 490 nm, through the phenol-sulfuric acid method as described by Connan [29] with minor modifications (0.25 mL of the sample was used instead of 0.5 mL). Determinations were made in triplicate, and the results were presented as g/L.

The produced sugars and the amount of biomass used in the hydrolysis were considered when calculating the biomass conversion yield (g sugar/100 g of dry algae).

## 3. Results and Discussion

According to the supplier, microalgae biomass is composed of around 30% (*w*/*w*) carbohydrates, which include neutral sugars, amino sugars and uronic acids [30].

### 3.1. Preliminary Results

The reaction conditions and results from the preliminary studies are presented in Table 2.

Under the studied conditions, acid concentration does not seem to have a clear effect on sugar concentration or yield, since similar results were obtained for the studied concentrations. Regarding hydrolysis time, this variable showed a positive impact on both sugar concentration and biomass conversion yield. Keeping constant the acid concentration (2.5% (*v*/*v*)) and the biomass/acid ratio (2.5% (*w*/*v*)), an increase of 19% was verified by increasing the reaction time from 30 to 90 min.

Results showed that the biomass/acid ratio has the opposite effect on sugar concentration and yield. As expected, an increase in the ratio (algae amount) resulted in an increase in sugar concentration, since there is a larger amount of biomass available for hydrolysis. However, it led to a decrease in biomass conversion yield. Namely, by increasing the ratio from 1% (*w*/*v*) to 10% (*w*/*v*), an increase of around 6-fold in sugar concentration and a decrease of 44% in yield was verified. In that sense, this parameter seems to be much more preponderant regarding the sugar concentration compared to the process yield. Furthermore, despite having a high yield, the use of 1% (*w*/*v*) biomass/acid ratio led to a very low sugar concentration. Since the final objective is bioethanol production, optimizing the sugar concentration for further fermentation is crucial. Even though the biomass conversion yield is an important parameter for the industrial feasibility of the process, using such a low biomass/acid ratio (1% (*w*/*v*)) seems unfeasible. Comparing the use of 2.5% (*w*/*v*) and 10% (*w*/*v*) biomass/acid ratio, the results show an increase in sugar concentration (by 270%) without a significant decrease in yield (7.5%).

Temperature is a relevant parameter for the dilute thermal acid hydrolysis of microalgae biomass. From the literature review, temperatures between 90 and 130 °C are usually employed in this process [6,9,21]. Therefore, two different temperatures (90/121 °C) were used in the preliminary assays. Higher hydrolysis temperature led to an increase of 22% in both sugar concentration and yield, keeping the other parameters constant. However, the increase in this variable can lead to a 46–90% rise in process costs [18], compromising the industrial feasibility of the process. Furthermore, Castro et al. [18] found the optimal hydrolysis temperature to be 80–90 °C. Additionally, the use of a thermoreactor led to a slight decrease both in sugar concentration and yield of around 5%, compared to the water bath assay for the same time, biomass/acid ratio and acid concentration. Taking into consideration the higher cost associated with a higher hydrolysis temperature, the differences were not considered relevant, so the use of the water bath was considered for the following studies.

The preliminary assays performed in the current study aimed to screen the most relevant variables for the hydrolysis of the biomass aiming further bioethanol production. The industrial and economic feasibility of the hydrolysis process are relevant factors to also take into consideration besides the yield to ensure the possibility of producing bioethanol from this biomass at a large scale. In that sense, the time, biomass/acid ratio and acid concentration were considered the most relevant parameters to optimization studies using the CCD.

Minh Thu et al. [6] optimized the hydrothermal acid pretreatment conditions for the microalgae *Chlamydomonas reinhardtii*, considering a biomass/acid ratio of 5% (*w*/*v*). Results show that longer hydrolysis times led to a higher sugar concentration and yield, which is in agreement with the present study. On the other hand, for an established hydrolysis time and temperature, higher acid concentrations increased the release of sugars (studied range: 1–5% (*v*/*v*)); however, for higher concentrations (3–5% (*v*/*v*)), the difference was much less significant. The optimal conditions obtained by the authors were 3% acid concentration for 30 min at 110 °C, which led to a sugar concentration of 28.5 g/L and a sugar conversion yield of 58 g/100 g. The optimal conditions the authors found were greater than those obtained in this study; however, the feedstock was different (different microalgae species), and the process temperature was higher. In the present study, only sulfuric acid concentrations over 2.5% (*v*/*v*) were studied, and no significant influence of acid concentration on sugar release was found.

Miranda et al. [17] studied the influence of acid concentration and biomass/acid ratio on the saccharification of *Scenedesmus obliquus* by autoclaving (120 °C). Even though a direct comparison of the result cannot be performed (different feedstocks and temperature), the biomass/acid ratio had a positive effect on the sugar concentration, which is in agreement with the verified result in the present study. Furthermore, for acid concentrations of 2.8% (*v*/*v*) and 5.6% (*v*/*v*), sugar yields of around 25 g/100 g and 27 g/100 g, respectively, were obtained. These findings corroborate the preliminary assays under analysis, where there were no significant differences in the sugar yield for acid concentrations between 2.5 and 4% (*v*/*v*).

Considering the obtained results, for the process optimization using the RSM design of experiments, the range of variables was defined as follows: the biomass/acid ratio was increased (5–15% (*w*/*v*)) to evaluate the point where mass transfer limitations start to occur, lowering the process yield, the acid concentration was lowered (0.5–3.5% (*v*/*v*)), to minimize the acid required since higher concentrations did not show a positive effect in the release of the sugars and a wider range of time (10–90 min) was used to determine whether a shorter time period would still lead to efficient hydrolysis.

### 3.2. Model Fitting

The results from the CCD are summarized in Table 3, where the responses obtained (sugar concentration and biomass conversion yield) are presented for the 32 performed assays.

Using the RSM, an empirical second-order polynomial model was obtained for the two response parameters. The regression models showed high fitting performance, R^2^ of 99.0% and 90.2% for the sugar concentration and yield, respectively. However, the results from the variance analysis (data not presented) showed that some of the variables had no statistical significance (*p* > 0.05), and, therefore, a reduced model with only statistically significant variables with a 95% confidence level (*p* < 0.05) was obtained through a stepwise algorithm, where the statistically non-significant variables were removed in descending order of *p*. For the yield model (*y*_2_), even though the amount of biomass (*x*_2_) was not a statistically significant variable (*p* = 0.126), it was kept in the reduced model for hierarchical reasons, as the quadratic term (*x*_2_^2^) and interaction with time (*x*_1_*∙x*_2_) were statistically significant. The empirical relationship between the responses (sugar concentration and yield) and the variables under study in the reduced model are presented in Equations (2) and (3).
(2)$$y_{1}=10.678+1{.196x}_{1}+5{.195x}_{2}+0{.433x}_{3}-0{.542x}_{2}^{2}+0{.268x}_{1}x_{2}+0{.412x}_{1}x_{3}+0{.405x}_{2}x_{3}$$
(3)$$y_{2}=10.662+1{.343x}_{1}+0{.185x}_{2}+0{.353x}_{3}-0{.642x}_{2}^{2}+0{.378x}_{1}x_{2}+0{.412x}_{1}x_{3}$$

The analyses of variance of the model for sugar concentration and yield are presented in the Appendix A (Appendix A, respectively). Regarding the sugar concentration (*y*_1_), the model was found to be statistically significant with a *p* lower than 0.001, demonstrating a maximum probability of 0.1% that the model can assume due to noise, validating the current model. In the same sense, the high coefficient of determination (R^2^ = 0.990) obtained shows that the model explains 99.0% of the variation in sugar concentration [20]. Finally, the high predicted R^2^ (0.981) indicates that the sugar concentration can be very well predicted by Equation (2). These results are also confirmed by the non-significant lack of fit (*p* = 0.565), which means that the data variation can be explained by the experimental error (pure error). In that sense, the model can accurately predict the sugar concentration in the studied variable ranges.

Similarly, the yield model was found to be statistically significant (*p* < 0.001) and to have an R^2^ value of 0.876. The model-predicted R^2^ value was 0.794, indicating that the model is expected to perform adequately in terms of prediction. The lack-of-fit test was not significant (*p* = 0.289), indicating that the model fits the data. Despite having a lower statistical significance than the sugar concentration model, the yield model can nonetheless properly adjust to experimental values and predict yield results.

Figure 1 shows the sugar concentration and process yield predicted by the model in relation to the obtained experimental values. Results show that the prediction capability of the models is relatively good, especially for the sugar concentration. Additionally, the residuals (distance of the points to the line) seem to have high homoscedasticity. Therefore, the models appear to be an accurate representation of the experimental values, which confirms the statistical results previously discussed.

In that sense, the models for sugar concentration and yield showed high statistical significance and strong predictive capability.

### 3.3. Surface Plots and Respective Analysis

Figure 2 shows the response surface plots for the models obtained for sugar concentration and yield response variables. Table 4 summarizes the β coefficients from Equations (2) and (3).

By analyzing Figure 2 and Table 4, it is possible to understand the behavior of the system in the studied parameter ranges.

#### 3.3.1. Sugar Concentration Model

Regarding the model for sugar concentration, the biomass/acid ratio appears to be the parameter that influences the response variable the most (β_2_ = 5.195). As expected, the amount of biomass has a positive effect on the sugar concentration, since there are more sugars in the system available to be hydrolyzed. In the same way, the linear influence of time seems to be relevant (β_1_ = 1.196). This variable shows a positive effect on the sugar concentration, as it results in an increase in contact time between the acid and the biomass at high temperatures. In that sense, there is a higher probability that the acid can catalyze the hydrolysis process. Regarding the linear effects, acid concentration is the parameter with less influence on the sugar concentration (β_3_ = 0.433). These results are in accordance with the literature, where a study from Miranda et al. [17] revealed that there is a positive effect of both biomass/acid ratio and acid concentration on the sugar concentration in the studied ranges.

It should be highlighted that some studies suggest that exposing the biomass to extreme acid and temperature conditions for long periods might degrade the hydrolyzed monosaccharides to other compounds not fermentable by yeast, compromising further bioethanol production [17]. In that sense, to avoid monosaccharide degradation and reduce production costs (sulfuric acid has a high cost [6]), using low acid concentrations in the hydrolysis might be an interesting approach from an industrial point of view.

Regarding the features of the JMP software, it was not possible to determine the optimal point for neither of the models, since a singular value was obtained. However, the maximum point for the studied range was determined with the optimization tool Microsoft Excel Solver. The maximum value for the sugar concentration model was found to be 18.05 g/L, obtained for *x*_1_, *x*_2_, *x*_3_ of 1, 1, 1 at the positive extreme of the variables, which corresponds to a hydrolysis time, biomass/acid ratio and acid concentration of 90 min, 15% (*w*/*v*), 3.5% (*v*/*v*). 

#### 3.3.2. Yield Model

The developed model for the process yield shows a different behavior compared to sugar concentration. Even though the linear effect of time (β_1_ = 1.343) is higher when compared to the acid concentration (β_3_ = 0.353)—confirming the relevance of lowering acid concentration instead of process time if milder hydrolysis conditions are required—the behavior of the system regarding the biomass/acid ratio is significantly different. The linear effect of this parameter is much less prominent (β_2_ = 0.185), and it is not statistically significant (*p* = 0.126), whereas the influence of the quadratic effect is clear (β_22_ = −0.642) in Figure 2b,f. This shows that the process yield increases with an increase in biomass/acid ratio until it reaches a point where mass transfer is compromised, and, even though there is more biomass in the system (therefore more sugars available to be hydrolyzed), the system in those conditions is not able to completely hydrolyze the microalgae polysaccharides.

These results are in accordance with the study of Ho et al. [2], where the sulfuric acid concentration showed a positive effect on the sugar yield. The study also reported a significant decrease in hydrolysis efficiency after reaching a point of mass transfer or reaction kinetics limitations. However, that point is reached at a biomass/acid ratio of 6% (*w*/*v*), a lower value compared to the current study, which can be justified by the different microalgae species used (*Scenedesmus obliquus*).

As occurred for the sugar concentration model, the maximum value for yield was obtained through Microsoft Excel Solver. The maximum value (12.86 g/100 g) was obtained for *x*_1_, *x*_2_, *x*_3_ of 1, −0.15, 1, corresponding to a hydrolysis time, biomass/acid ratio and acid concentration of 90 min, 9.3% (*w*/*v*) and 3.5% (*v*/*v*).

#### 3.3.3. Model Validation

To validate the model prediction, laboratory experiments were performed in conditions close to the optimal yield conditions, and the obtained experimental results are presented in Table 5. The low error (<15%) obtained for both models confirms the adequate prediction capacity of the models.

The sugar concentration and yield models indicate the biomass/acid ratio and time as key variables, respectively. Optimal conditions were determined, and finally, model validation confirmed the models’ predictive capability.

### 3.4. Critical Analysis

Even though it was impossible to obtain an optimal value in the studied variables intervals, studying points outside the defined ranges was considered of low relevance for further research developments. Regarding the process time, increasing this variable indefinitely, despite potentially increasing sugar concentration and yield, results in a decrease in productivity and an increase in energy costs (to maintain the reactor at the operational temperature of 90 °C). In that sense, the literature concerning thermal acid hydrolysis of microalgae uses process times in the studied range [2,17]. Higher temperatures might be unfeasible due to economic constraints, and temperatures lower than 90 °C are known as generally ineffective; however, the evaluation of this variable remains relevant for future studies. Furthermore, long residence times at high temperatures can be associated with the degradation of glucose [6], which would decrease the amount of sugars available for fermentation, although that was not possible to detect in this study (total sugars quantification was employed). Focusing on the acid concentration, despite verifying a positive effect of this variable on the studied responses, high acid concentrations are usually associated with yeast growth inhibition in the fermentation process after neutralization. For instance, Markou et al. [31] found that using H_2_SO_4_ at 7% (*v*/*v*) inhibits yeast growth and metabolism, further compromising bioethanol production. Finally, although high biomass/acid ratios result in high sugar concentrations, there is a decrease in process yield, most likely due to mass transfer limitations that occur for high solid contents [2]. In that sense, increasing this variable outside the studied range would compromise the process yield and the economic feasibility of the process.

No studies in the literature were found concerning the optimization of saccharification of microalgae through thermal acid hydrolysis using RSM. Thus, no direct comparison with the literature can be carried out. However, Castro et al. [18] performed a full factorial optimization of the process and found that the optimal conditions were 10% (*w*/*v*) biomass/acid ratio, 2.8% (*v*/*v*) H_2_SO_4_ concentration, 120 min process time and 80–90 °C temperature. In these conditions, a sugar yield of 16.6 g/100 g was found. Although using different species and methodologies, the authors obtained similar optimal conditions to those obtained in the present study (90 min process time; 9.3% (*w*/*v*) biomass/acid ratio; 3.5% (*v*/*v*) H_2_SO_4_ concentration; 90 °C), also resulting in a similar yield (12.86 g/100 g). In addition, the results indicate that the *Aurantiochytrium* sp. species requires a similar temperature to hydrolyze the sugars to the mixture of microalgae species used in the mentioned study (*Scenedesmus*, *Chlorella*, *Ankistrosdemus*, *Micromonas* and *Chlamydomonas*).

The production costs are one of the major hurdles for large-scale bioethanol production from microalgae [11]. Considering the economic feasibility of the process, a biorefinery approach seems to be interesting to explore for this kind of biomass [11], as it can obtain several compounds with industrial and economic value. More specifically, pigments, lipids and other compounds can be successively extracted, and the remaining biomass (rich in carbohydrates) can be used for renewable energy production, namely through bioethanol production. As an example, *Aurantiochytrium* sp. biomass is rich in docosahexaenoic acid (DHA), an omega-3 lipid with high value for human health [32], which can be extracted, and the remaining biomass can be further processed as a feedstock for bioethanol production.

A techno-economic analysis of the studied process was not performed; however, dilute thermal acid hydrolysis is widely reported as a low-cost method [33] compared to using enzymes. In fact, enzymatic hydrolysis is usually less cost-effective (due to high enzyme cost and low process velocity [34,35]), which could be a disadvantage at the industrial scale, and it has been reported that the enzyme can comprise about 20% of the total cost of bioethanol production [36]. In that sense, it is expected that the studied process represents a low-cost method for the saccharification of microalgae biomass. Furthermore, the valorization of other biomass compounds following a biorefinery approach can be interesting from an economic point of view.

## 4. Conclusions

The thermal acid hydrolysis of microalgae biomass to release sugars using sulfuric acid was assessed. Response surface methodology was employed to model the process and understand the impact of relevant parameters (time; biomass/acid ratio; acid concentration) on sugar concentration and process yield. A statistically significant quadratic model was obtained for each response variable (sugar concentration: R^2^ = 0.990; yield: R^2^ = 0.876). Regarding sugar concentration, biomass/acid ratio was the variable with the most influence in the system, whereas, for yield, time was the most influential parameter. Furthermore, the conditions that maximized the responses in the studied ranges were found to be 90 min, 15% (*w*/*v*) biomass/acid ratio, 3.5% (*v*/*v*) acid concentration 90 min and 9.3% (*w*/*v*) biomass/acid ratio and 3.5% (*v*/*v*) acid concentration for sugar concentration and yield, respectively. Further studies should be conducted to evaluate bioethanol production under the obtained optimal hydrolysis conditions and its integration with a biorefinery approach.

## Figures and Tables

**Figure 1 biology-12-00935-f001:**
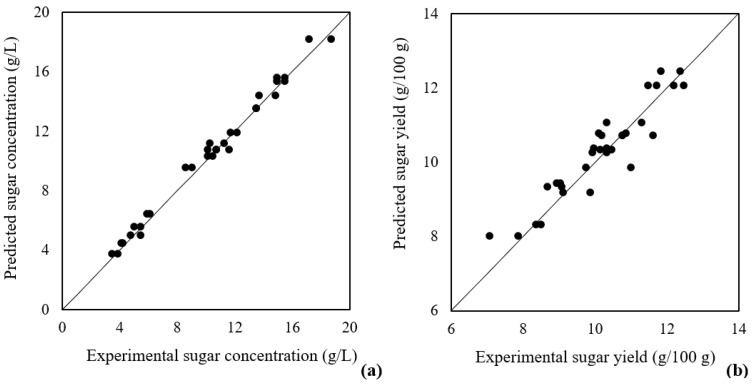
Predicted sugar concentration (**a**) (R^2^ = 0.990) and yield (**b**) (R^2^ = 0.876) versus the experimental values obtained in the same conditions. Line corresponds to the absence of error.

**Figure 2 biology-12-00935-f002:**
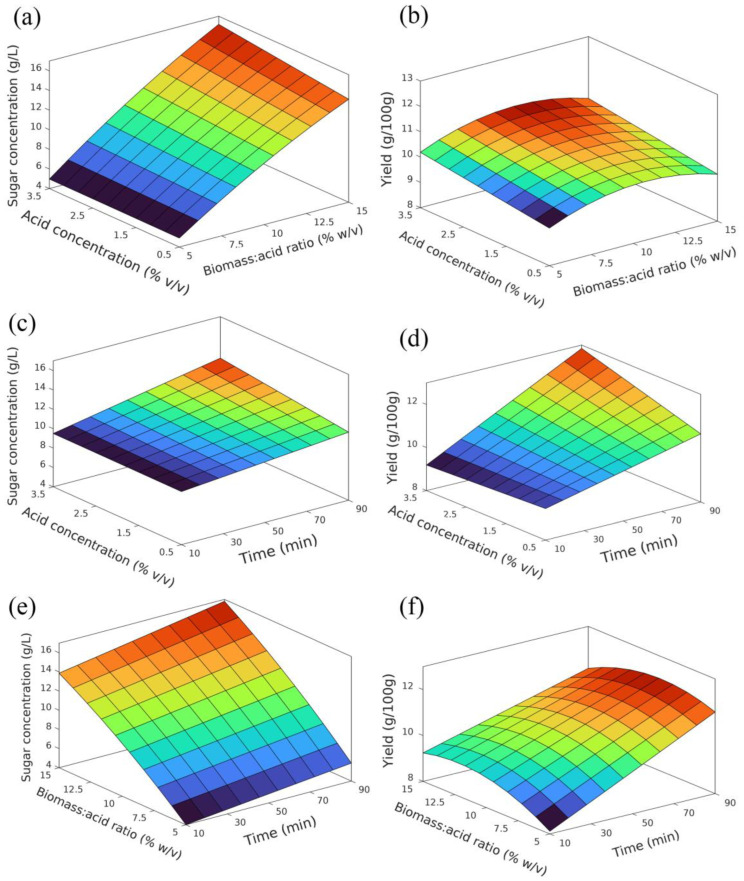
Response surface for sugar concentration (**a**,**c**,**e**) and yield (**b**,**d**,**f**): (**a**,**b**) 50 min hydrolysis time; (**c**,**d**) 10% (*w*/*v*) biomass/acid ratio; (**e**,**f**) 2.0% (*v*/*v*) acid concentration.

**Table 1 biology-12-00935-t001:** Range of studied variables and code levels used in the central composite design.

Parameter		Unit	Range and Level		
			−1	0	1
*x* _1_	Time	min	10	50	90
*x* _2_	Biomass/acid ratio	% (*w*/*v*)	5	10	15
*x* _3_	Acid concentration	% (*v*/*v*)	0.5	2	3.5

**Table 2 biology-12-00935-t002:** Preliminary assays performed for different durations (30–90 min), biomass/acid ratio (1–10% (*w*/*v*)) and acid concentrations (2.5–4% (*v*/*v*)). Assays at 90 °C and 121 °C were performed in a water bath and autoclave, respectively.

Temperature (°C)	[H_2_SO_4_] (% (*v*/*v*))	Time (min)	Biomass/Acid Ratio (% (*w*/*v*))	Yield (g/100 g)	Sugar Concentration (g/L)
90	2.5	30	2.5	13 ± 1	3.2 ± 0.3
	2.5	60	2.5	13.2 ± 0.6	3.3 ± 0.1
	2.5	90	2.5	14 ± 1	3.6 ± 0.4
	2.5	90	1	24 ± 3	2.4 ± 0.3
	2.5	90	10	13.3 ± 0.7	13.3 ± 0.7
	3	90	2.5	13.7 ± 0.2	3.48 ± 0.05
	4	90	2.5	14 ± 1	3.7 ± 0.4
121	2.5	90	2.5	18 ± 1	4.6 ± 0.3
100–148 *	3	90	2.5	13.2 ± 0.7	3.3 ± 0.2

Note: Experimental values are presented as mean value ± standard deviation. * 100–148 °C corresponds to the temperature of the thermoreactor, considering 100 °C for 60 min plus 148 °C for 30 min.

**Table 3 biology-12-00935-t003:** Central composite design and the corresponding responses (sugar concentration—*y*_1_; yield—*y*_2_).

Run.	Time (min)	Biomass/Acid Ratio (% (*w*/*v*))	[H_2_SO_4_] (% (*v*/*v*))	*x* _1_	*x* _2_	*x* _3_	*y*_1_ (g/L)	*y*_2_ (g/100 g)
1	50	10	0.5	0	0	−1	10.55	10.52
2	50	5	2	0	−1	0	5.54	11.05
3	50	15	2	0	1	0	15.57	10.38
4	10	5	0.5	−1	−1	−1	4.32	8.57
5	10	10	2	−1	0	0	9.13	9.12
6	10	15	3.5	−1	1	1	14.92	9.95
7	90	5	0.5	1	−1	−1	5.11	10.16
8	10	10	2	−1	0	0	8.72	8.72
9	10	5	3.5	−1	−1	1	3.98	7.92
10	90	5	0.5	1	−1	−1	5.52	10.92
11	10	15	0.5	−1	1	−1	13.63	9.07
12	10	5	0.5	−1	−1	−1	4.24	8.44
13	90	10	2	1	0	0	11.78	11.76
14	50	5	2	0	−1	0	4.91	9.80
15	90	15	3.5	1	1	1	18.89	12.54
16	50	10	2	0	0	0	10.84	10.82
17	90	15	3.5	1	1	1	17.32	11.53
18	10	5	3.5	−1	−1	1	3.59	7.14
19	50	10	0.5	0	0	−1	10.23	10.22
20	90	15	0.5	1	1	−1	15.08	10.05
21	50	10	2	0	0	0	10.85	10.83
22	90	5	3.5	1	−1	1	5.98	11.92
23	90	5	3.5	1	−1	1	6.23	12.43
24	10	15	3.5	−1	1	1	13.77	9.18
25	50	10	2	0	0	0	10.28	10.26
26	10	15	0.5	−1	1	−1	13.54	8.98
27	50	10	2	0	0	0	11.70	11.68
28	50	15	2	0	1	0	15.02	10.00
29	50	10	3.5	0	0	1	10.41	10.39
30	50	10	3.5	0	0	1	11.39	11.36
31	90	15	0.5	1	1	−1	15.59	10.38
32	90	10	2	1	0	0	12.26	12.26

Note: Design of experiments considers two replicates for each condition and four replicates for central point (0, 0, 0).

**Table 4 biology-12-00935-t004:** Results concerning β coefficients for the reduced sugar concentration and yield models.

β Coefficients	Model	
	Yield	Sugar Concentration
β_0_	10.662	10.678
β_1_	1.343	1.196
β_2_	0.185 *	5.195
β_3_	0.353	0.433
β_12_	0.378	0.268
β_13_	0.412	0.412
β_23_	-	0.405
β_2_^2^	−0.642	−0.542

* Statistically insignificant parameter at a 95% confidence level, kept in the reduced model for hierarchical reasons.

**Table 5 biology-12-00935-t005:** Results for the validation essay and differences between the prediction and experimental results.

	[H_2_SO_4_] (% (*v*/*v*))	Time (min)	Biomass/Acid Ratio (% (*w*/*v*))	Yield (g/100 g)	Sugar Concentration (g/L)
Prediction Experimental	3.5	90	10	12.84 11.41 ± 0.04	12.72 11.45 ± 0.03
Prediction error (%)				≈11	≈13

Note: The experimental values are presented as average ± error (two replicates, expressed by the absolute difference between measured values and average). Prediction error was calculated by the relative difference (%) of the prediction relative to the experimental value.

## Data Availability

Not applicable.

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
