# Peer review of "Microalgae to Bioenergy: Optimization of Aurantiochytrium sp. Saccharification"

_biology, 2023, doi:10.3390/biology12070935_

Round 1

Reviewer 1 Report

The paper entitled ''Optimisation of thermal acid hydrolysis of Aurantiochytrium sp.'' aims to investigate the optimization of the critical parameters for the thermal acid hydrolysis of microalgae cells. The parameters, including acid concentration, hydrolysis time, and biomass:acid ratio were optimized using response surface methodology for further bioethanol production. My comments are as follows.

1.     The manuscript title needs to be revised since it should be attractive to potential readers.

2.     Materials and Methods, Section 2.2: Please check the sentence in line 126.

3.     Materials and Methods: The author should explain why the central composite design was selected.

4.     Materials and Methods: Please give information about the statistical analysis done to validate the model results.

5.     Add a summarizing sentence on what should be remembered after each section in the results section.

6.     Figure 1: R2 values need to be to graphs.

7.     The authors should give references regarding the analyses, such as sugar quantification.

8.     More references should be added to the manuscript.

Author Response

Reviewer #1 Comments

Answer

The manuscript title needs to be revised since it should be attractive to potential readers.

The authors thank the reviewer's comment. The title was changed to “Optimisation of microalgae Aurantiochytrium sp. sacharification aiming bioethanol production” in order to be attractive for readers in the energy field.

Materials and Methods, Section 2.2: Please check the sentence in line 126.

In order to respond to the reviewer comment, the information was clarified in the Materials and Methods section (line 126).

Materials and Methods: The author should explain why the central composite design was selected.

The authors thank the reviewer's comment. The rationale on why central composite design was selected for the design of experiments was clarified in the Materials and Methods section (line 129-130).

Materials and Methods: Please give information about the statistical analysis done to validate the model results.

In order to respond to the reviewer comment, additional information about the statistical analysis were added to the Materials and Methods section. Changes are highlighted in lines 142-145.

Add a summarizing sentence on what should be

remembered after each section in the results section.

In order to respond to the reviewer comment, a summarizing sentence was added in the end of each results sub-section.

Figure 1: R2 values need to be to graphs.

The authors thank the reviewer's comment. The R2 values were added to the caption of Figure 1.

The authors should give references regarding the analyses, such as sugar quantification.

The authors thank the reviewer's comment. In order to clarify the method used for sugar quantification, the manuscript was revised in lines 154-159. The reference corresponding to the method is presented in line 156.

More references should be added to the manuscript.

Additional references were added in the manuscript to support the discussion of results. Even though new efforts were made to find additional studies that used RSM for the optimisation of acid hydrolysis conditions, no further studies were found to make direct comparisons with the obtained results.

Reviewer 2 Report

This article is relatively conventional and reports a design of experiment methodology ( CCD) strategy for optimizing thermal acid hydrolysis for obtaining the best condition for saccharification of Aurantiochytrium sp. biomass. Before publication, the article can be improved with some remarks and suggestions:

Page 2, line 127. Please add more details about the software used (producer, country… )

Table 2. Please uniformize the data in the table using two decimals.

For 3.3.1 and 3.2.2 paragraphs, can authors expand the RSM plots interpretations, please?

Page 9. I suggest specifying in a table the beta coefficients.

Page 10. Please explain if this study's optimized thermal acid hydrolysis can be financially viable. 

Author Response

Reviewer #2 Comments

Answer

Page 2, line 127. Please add more details about the software used (producer, country… )

In order to respond to the reviewer comment, the details were added in the manuscript (line 128-129).

Table 2. Please uniformize the data in the table using two decimals.

The authors thank the reviewer's comment. The results in Table 2 were presented according to the number of significant figures of the error (presenting the decimal places according to the value with significance considering the error). For that reason, the number of decimals are not the same for all cases. The authors propose to maintain the current presentation of results, being however available for changes if the reviewer considers that another presentation is preferred.  

For 3.3.1 and 3.2.2 paragraphs, can authors expand the RSM plots interpretations, please?

In order to respond to the reviewer comment, a more comprehensive view on the impact of the variable on each of the responses was provided in the reviewed manuscript (lines 287-296; 324-329).

Page 9. I suggest specifying in a table the beta coefficients.

The authors thank the reviewer's comment. A table with the beta coefficients was added in Section 3.3 (Table 4).

Page 10. Please explain if this study's optimized thermal acid hydrolysis can be financially viable.

The authors thank the reviewer's comment. An explanation concerning the financial aspects of the hydrolysis was added in the end of Section 3.4 (lines 386-391).

Round 2

Reviewer 1 Report

I have carefully reviewed the R1 version of the manuscript. I do not think the R1 version is suitable for publication in Biology. I believe that the manuscript still needs major revisions that the authors must carefully tackle. In addition to some previous comments, I have made some additional comments. I kindly suggest that the authors consider my comments and incorporate them into the manuscript accordingly. My comments are as follows.

1.   The title should summarize the key points and provide a clear indication of the topic that is discussed in the article. A good title should be short and to the point yet still provide enough information to give readers an idea of what the article is about. The title should also be specific enough to attract the target readers and interesting enough to encourage readers to read the article. It still does not look attractive. Please consider improving it.

2.   Materials and Methods: The author should explain why the central composite design was selected. This is not a convincing response (Lines 129-130). Have the authors evaluated other designs? There are several types of response surface designs available in statistics, including Box-Behnken design (BBD), Doehlert design (D-optimal), Face-centered central composite design (FCC), Rotatable central composite design (RCC), Hybrid design (mix of CCD and BBD). Maybe other designs show better fitting with the experimental results?

3.   Materials and Methods: Please give information about the statistical analysis done to validate the model results. I could not find satisfactory information about the statistical analysis in lines 142-145. Please give more details.

4.   Did the authors investigate the optimal hydrolysis temperature? The hydrolysis process temperature for H2SO4 is crucial because it can affect the efficiency and selectivity of the hydrolysis reaction. The hydrolysis of carbohydrates using H2SO4 is typically performed at high temperatures, which can break down the complex carbohydrates into simpler sugars. However, if the temperature is too high, it can lead to over-hydrolysis and degradation of the desired product. On the other hand, if the temperature is too low, it can result in incomplete hydrolysis and lower yields of the desired product.

5.   Please mention the cultivation conditions of microalgae cells because cultivation conditions of microalgae cells can significantly affect the carbohydrate content of the cells.

6.   Figure 2: The font size of the titles of the X and Y axes needs to be increased. They must be readable.

 Minor editing of English language required

Author Response

Porto, 9th June 2023

Subject: Revision of the manuscript “Microalgae to bioenergy: optimization of Aurantiochytrium sp. saccharification”

Thank you for the opportunity to revise the manuscript.

The comments of the reviewer and respective answers are presented in the following table.

Reviewer #1 Comments

Answer

The title should summarize the key points and provide a clear indication of the topic that is discussed in the article. A good title should be short and to the point yet still provide enough information to give readers an idea of what the article is about. The title should also be specific enough to attract the target readers and interesting enough to encourage readers to read the article. It still does not look attractive. Please consider improving it.

The authors thank the reviewer's comment. The title was changed to “Microalgae to bioenergy: optimization of Aurantiochytrium sp. saccharification” in order to be attractive, short and to the point.

Another longer alternative that provides more information would be “Thermal acid hydrolysis of microalgae aiming bioethanol production: optimisation of processing time, biomass:acid ratio and acid concentration”. This alternative can be used if the reviewer thinks it is more appropriate.

Materials and Methods: The author should explain why the central composite design was selected. This is not a convincing response (Lines 129-130). Have the authors evaluated other designs? There are several types of response surface designs available in statistics, including Box-Behnken design (BBD), Doehlert design (D-optimal), central composite design (CCD), Hybrid design (mix of CCD and BBD). Maybe other designs show better fitting with the experimental results?

The authors thank the reviewer’s comment. The CCD is one the most widely used RSM design of experiments. In the current work, a 3-factor design is employed so only the use of standard designs (BBD and CCD) were considered. The use of complex designs such as hybrid design and D-optimal were considered inappropriate for the study. Moreover, the comparison of these RSM designs lead to the increase of the number of planed experiments, due to the different criteria for the design of experiments.

The rationale behind the decision on using CCD over BBD is discussed in lines 133-140.

Materials and Methods: Please give information about the statistical analysis done to validate the model results. I could not find satisfactory information about the statistical analysis in lines 142-145. Please give more details.

According to the reviewer’s comment, a more complete explanation regarding the statistical analysis done to validate the model results is provided in lines 152-167 and 172-176.

Did the authors investigate the optimal hydrolysis temperature? The hydrolysis process temperature for H2SO4 is crucial because it can affect the efficiency and selectivity of the hydrolysis reaction. The hydrolysis of carbohydrates using H2SO4 is typically performed at high temperatures, which can break down the complex carbohydrates into simpler sugars. However, if the temperature is too high, it can lead to over-hydrolysis and degradation of the desired product. On the other hand, if the temperature is too low, it can result in incomplete hydrolysis and lower yields of the desired product.

The authors thank the reviewer’s comment. The effect of temperature was not explored in the current work. However, the hydrolysis temperature took into consideration the optimized temperature of the work from Castro et al. (lines 391-394) that found the optimal hydrolysis temperature to be 80-90 ºC. Furthermore, as explained in lines 224-226, higher hydrolysis temperatures significantly increase energy costs and therefore it was considered of low relevance to be explored in the current study.

No studies were found regarding thermal acid hydrolysis of microalgae biomass using temperatures lower than 80 ºC since it would decrease the yield of the process, also according to the reviewer’s comment.

In that sense, using 90 ºC as hydrolysis temperature was considered the most appropriate for the current study.

Such information was clarified in the revised version of the manuscript in lines 126-128.

Please mention the cultivation conditions of microalgae cells because cultivation conditions of microalgae cells can significantly affect the carbohydrate content of the cells.

Following the reviewer’s comment, the cultivation conditions of microalgae cells are presented in lines 116-119.

The medium composition cannot unfortunately be provided due to the fact that, according to the supplier, it is confidential.

Figure 2: The font size of the titles of the X and Y axes needs to be increased. They must be readable.

According to the reviewer’s comment, the font size of the titles of the X and Y axes in Figure 2 were increased.

With kind regards,

José Carlos Pires

Round 3

Reviewer 1 Report

After careful review, my opinion is that this manuscript does not meet the publication requirements as it contains a serious flaw, so it is not suitable for publication in Biology. The authors did not perform a screening analysis before RSM. Screening analysis is used to identify the most important inputs that will affect the output. These inputs are then used in the RSM analysis to identify the most important factors that influence the output. By identifying the most important factors, RSM can be used to optimize the process and find the best combination of inputs to achieve the desired output. Screening analysis helps to reduce the number of factors that need to be considered in the RSM analysis. This reduces the amount of time and cost associated with the analysis and ensures that the most important factors are considered. It also helps to identify which inputs are not important and can be excluded from the RSM analysis. Screening analysis also helps to identify potential interactions between inputs. This can help to identify any interactions that may influence the output and can be used to reduce the complexity of the RSM analysis. Overall, screening analysis is a critical step before undertaking RSM. It helps to identify the most important inputs, reduce the complexity of the analysis, and identify potential interactions between inputs. By performing screening analysis, RSM can be used to optimize the process and find the best combination of inputs to achieve the desired output.

 In this study, the authors did not include in hydrolysis temperature in the design, which is considered a serious flaw. Because it can affect the efficiency and selectivity of the hydrolysis reaction. The hydrolysis of carbohydrates using H2SO4 is typically performed at high temperatures, which can break down the complex carbohydrates into simpler sugars. However, if the temperature is too high, it can lead to over-hydrolysis and degradation of the desired product. On the other hand, if the temperature is too low, it can result in incomplete hydrolysis and lower yields of the desired product. In this context, this parameter should have been included in the design as a critical parameter.

As a result, I have identified additional experiments that are needed to support the result, and I have concerns about the overall accuracy and validity of the research. Furthermore, I have determined that the research was not conducted correctly, and there are significant issues with your data collection and analysis. These concerns are too significant to overlook, and I believe that they undermine the credibility of the findings. Considering these points, the manuscript does not merit publication in Biology.

Moderate editing of English language required

Author Response

Porto, 22nd June 2023

Subject: Revision of the manuscript “Microalgae to bioenergy: optimization of Aurantiochytrium sp. saccharification”.

Thank you for the opportunity to revise the manuscript. The comments of the reviewer and respective answers are presented in the following text.

In the present revision, the authors have made an effort to address as much as possible the concerns of the reviewer.

The major reviewer comments were that “The authors did not perform a screening analysis before RSM” and that “the authors did not include in hydrolysis temperature in the design”.

The authors agree that the screening analysis is a very important aspect. In fact, sections 2.1/3.1 describe the methodology and results of “preliminary studies to define variables range”, which clearly shows that a screening was conducted. In those preliminary assays, the temperature was also studied but such results were not included in the previous version of the manuscript since the variable was not further considered for the RSM design.

In order to complement the study and to respond to the reviewer comment, the preliminary assays considering the influence of the temperature in sugars release were now added to the manuscript, being the results discussed accordingly. The rationale behind the decision of using 90 ºC as the hydrolysis temperature for all the experiments in the design is now clarified in the revised version of the manuscript.

The revised manuscript consequently includes: revisions in the end of the paragraph of section 2.1 Preliminary studies: definition of variables range regarding conditions of preliminary studies and inclusion of temperature (lines 123-129); First paragraph of section 2.2 Experimental design, explaining the selection of the parameters and the use of RSM (lines 133-136); Table 2 (pg. 5) has included the results at different temperatures and second and third paragraphs of pg. 6 include discussion regarding the influence of temperature (lines 220-239); finally, in the critical analysis the relevance of the temperature in future studies was analysed (lines 402-405).

In fact, the aim of the screening analysis performed in the preliminary assays was to determine the most important factors to be considered in the experimental design, keeping in mind the minimization of “the amount of time and cost associated with the analysis”, as stated by the reviewer. In that sense, even though the temperature is a relevant parameter for the hydrolysis of the biomass, considering the results obtained it was not considered for the current design.

In the answer to the major comments by the reviewer the authors have also responded to the editor suggestions, namely including clear explanations on why "the authors did not include hydrolysis temperature in the design" as well as its importance and proposal for future studies (lines 123-129; 133-136; 220-239; 402-405).

With kind regards,

José Carlos Pires
